# Ferroptosis-Related Gene SLC1A5 Is a Novel Prognostic Biomarker and Correlates with Immune Microenvironment in HBV-Related HCC

**DOI:** 10.3390/jcm12051715

**Published:** 2023-02-21

**Authors:** Hanwen Su, Youyi Liu, Jingtao Huang

**Affiliations:** 1Department of Clinical Laboratory, Institute of Translational Medicine, Renmin Hospital of Wuhan University, Wuhan 430060, China; 2Wuxi School of Medicine, Jiangnan University, Wuxi 214122, China

**Keywords:** HBV, HCC, SLC1A5, ferroptosis-related gene, tumor microenvironment

## Abstract

(1) Background: Hepatocellular carcinoma (HCC) is one of the most common cancers worldwide with limited treatment satisfaction. Finding new therapeutic targets has remained a major challenge. Ferroptosis is an iron-dependent cell death program that plays a regulatory role in HBV infection and HCC development. It is necessary to classify the roles of ferroptosis or ferroptosis-related genes (FRGs) in HBV-related HCC progression. (2) Methods: We conducted a matched case–control study from the TCGA database, retrospectively collecting demographic data and common clinical indicators from all subjects. The Kaplan–Meier curve, univariate and multivariate cox regression analysis of the FRGs were used to explore the risk factors for HBV-related HCC. The CIBERSORT algorithm and TIDE algorithm were executed to evaluate the functions of FRGs in the tumor-immune environment. (3) Results: A total of 145 HBV-positive HCC patients and 266 HBV-negative HCC patients were enrolled in our study. Four ferroptosis related genes (FANCD2, CS, CISD1 and SLC1A5) were positively correlated with the progression of HBV-related HCC. Among them, SLC1A5 was an independent risk factor for HBV-related HCC, and correlated with poor prognosis, advanced progression and an immunosuppression microenvironment. (4) Conclusions: Here, we revealed that a ferroptosis-related gene, SLC1A5, may be an excellent predictor of HBV-related HCC and may provide insight into the development of innovative possible therapeutic techniques.

## 1. Introduction

Liver cancer is one of the most common cancers and represents a major global health challenge [1]. Primary liver cancer is the sixth most commonly diagnosed cancer and has the third highest mortality, with almost 906,000 new cases and 830,000 deaths annually [2]. Hepatocellular carcinoma (HCC) is the major form of liver cancer, accounting for ~90% of cases, with a relative 5-year survival rate of 18% [1,3]. The chronic infections with the hepatitis B virus (HBV) or hepatitis C virus (HCV) are the most prominent risk for the development of HCC [4]. Non-alcoholic steatohepatitis (NASH), which is associated with metabolic syndrome or diabetes, is becoming a growing cause of HCC in the West [5]. Additionally, alcohol abuse, obesity, nicotine use and exposure to aflatoxin B1 are also associated with increased incidence of HCC [6]. The prevalence of risk factors for HCC depends on the geographical area, with a predominance of HBV in Asia, HCV in Japan, and NASH in Europe and North America [1]. As a major aetiology of HCC, HBV contributes to the development of HCC through various mechanisms, such as interference with signaling pathways, genome integration and influence on genomic instability [7]. To verify the pathogenesis of HBV-related HCC, it is important to identify the host biology factors of HBV infection.

Ferroptosis is an iron-dependent cell death program, which is distinct from apoptosis, necrosis and autophagy, with the primary feature being the accumulation of reactive oxygen species (ROS) and lipid peroxides (LPOs) [8]. Ferroptosis has been shown to play an important role in the progression of carcinogenesis and may be used as a potential novel strategy for cancer treatment [9]. Meanwhile, recent studies have discovered that the liver is predisposed to oxidative injury and excessive iron accumulation is a major characteristic of liver diseases [10]. Therefore, ferroptosis has been implicated in the progression of HBV infection and HCC development, and can be the potential therapeutic target for HCC [11]. For example, sorafenib, a first-line drug for HCC, has been shown to improve survival and induce ferroptosis via altering the TP53 and Rb signaling pathways [12]. In addition, ferroptosis and hepatotoxicity were potentiated by HBV X protein (HBx) via suppressing SLC7A11 through H3K27me3 modification by EZH2 in acute liver failure in mice models [13]. SLC1A5, also known as alanine serine cysteine-preferring transporter 2 (ASCT2), transporting glutamine in a Na^+^-dependent manner, was found to be an important ferroptosis modulater [14]. Via the modulation of SLC1A5 and the mTORC1 signaling pathway, the discoidin domain receptor 1 promotes the progression of hepatocellular carcinoma [15]. Meanwhile, researchers discovered that ferroptosis was involved in the immunosuppressive tumor microenvironment (TME) and γδ T-cell imbalance in HCC [16]. Although many mechanisms of ferroptosis have been discovered in cancers, few studies focus on its role in HBV-related HCC [17]. Therefore, it is important to identify specific biomarkers of ferroptosis involved in the regulation of tumor pathogenesis in HBV-related HCC.

In this research, we identified 10 distinct ferroptosis-related genes (FRGs) in HBV infection based on the expression data from the TCGA database. Additionally, the FRG SLC1A5 was discovered as an independent risk factor in HBV-related HCC and was associated with a poor prognosis. In addition, the TIDE score of SLC1A5 was constructed to quantify the immune microenvironment and immune escape probability of HBV-related HCC patients and to predict the response to immunotherapy. Collectively, SLC1A5 may be an excellent predictor of HBV-related HCC and may provide new avenues for the development of innovative possible therapeutic techniques.

## 2. Materials and Methods

### 2.1. Sample Data Acquisition and HBV Infection-Associated FRGs Identification

RNA-sequencing expression profiles and corresponding clinical information for HCC were downloaded from the TCGA dataset (https://portal.gdc.cancer.gov/ (accessed on 10 February 2023)), containing 145 HBV-positive HCC samples and 266 HBV-negative HCC samples. FerrDb (http://www.zhounan.org/ferrdb (accessed on 10 February 2023)) is reported to be the first repository of ferroptosis modulators and indicators, as well as ferroptosis-disease connections, which was manually collated [18]. A total of 24 FRGs obtained from FerrDb were analyzed in this study, drawing on the research of Liu et al. [19]. The expression levels of the 24 FRGs were compared between HBV positive and negative HCC samples. The FRGs significantly up-regulated in HBV-positive samples were screened and then analyzed in HBV-related HCC patients with different TNM stages to identify the HBV infection-associated FRGs.

### 2.2. Survival Analysis of the FRGs

The Kaplan–Meier curve was used to display the overall survival (OS) and disease specific survival (DSS) of the samples, and the logarithmic rank test was utilized to determine the statistical difference. Hazard ratio (HR) with 95% confidence interval (CI) was generated by logarithmic rank test and univariate cox proportional hazards regression. The “timeROC” package of R was applied to generate the receiver operating characteristic (ROC) curve, and the prediction accuracy of the genes was examined by calculating the area under the curve (AUC) of one-, three-, and five-year OS. All analysis methods and R packages were implemented by R (foundation for statistical computing 2020) version 4.0.3. A *p* value of <0.05 was considered statistically significant.

### 2.3. Evaluation of the Prognostic Value of the FRGs

According to the RNA-sequencing expression profiles and corresponding clinical information for HBV-related HCC downloaded from the TCGA, univariate and multivariate cox regression analysis of the FRGs and clinicopathological features were performed to identify the proper terms to build the nomogram. The forest was used to show the *p* value, HR and 95% CI of each variable through ‘forestplot’ R package. A nomogram was developed based on the results of multivariate cox proportional hazards analysis to predict the 3-year overall recurrence. The nomogram provided a graphical representation of the factors which can be used to calculate the risk of recurrence for an individual patient by the points associated with each risk factor using the ’rms‘ R package.

### 2.4. Computation of Immune Cellular Fraction and Prediction of Response to ICB

To assess the reliable results of immune score evaluation, an R software package “immuneeconv” was used and the relative abundance of 22 different immune cells in distinct groups using the CIBERSORT algorithm [20]. The R software “ggstatsplot” package was used to plot the correlations between gene expression and immune score. Spearman’s correlation analysis was used to describe the correlation between quantitative variables without a normal distribution. A *p* value of < 0.05 was considered statistically significant.

Eight genes, SIGLEC15, TIGIT, CD274, HAVCR2, PDCD1, CTLA4, LAG3 and PDCD1LG2, were selected to be immune-checkpoint–relevant transcripts, and their expression values in groups were extracted. The Tumor Immune Dysfunction and Exclusion (TIDE, http://tide.dfci.harvard.edu/ (accessed on 10 February 2023) algorithm was performed to predict the potential ICB response between the low- and high-expression groups. All the above analysis methods and R packages were implemented by R foundation for statistical computing (2020) version 4.0.3.

### 2.5. Analysis of the Correlations between Gene Expression and Signaling Pathways

We collected 105 common signaling pathways and corresponding genes contained in the Gene Set Enrichment Analysis (GSEA) database. R software “GSVA” package was used to analyze, choosing parameter as method = “ssgsea”. The correlation between genes and pathway scores was analyzed using Spearman correlation. All the analysis methods and R packages were implemented by R version 4.0.3. A *p* value of <0.05 was considered statistically significant.

### 2.6. Statistical Analysis

All statistical analyses and data visualization were conducted by R version 4.0.3. All calculated *p* values were two-tailed and a *p* value of <0.05 was considered significant.

## 3. Results

### 3.1. Screening of Ferroptosis Related Genes Associated with HBV Infection in HCC

Based on the TCGA database, the statistical data of 145 HBV-positive HCC patients and 266 HBV-negative HCC patients were obtained. The mRNA expression levels of 24 ferroptosis-related genes (FRGs) were analyzed. The results demonstrated that 10 FRGs were significantly up-regulated in HBV-positive HCC patients compared with the HBV negative HCC patients (Figure 1A). Of these, 4 genes (FANCD2, CS, CISD1 and SLC1A5) were positively correlated with the progression of HBV-related HCC (Figure 1B) and were defined as HBV infection-associated FRGs.

### 3.2. Survival Analysis of the 4 HBV Infection-Associated FRGs in HBV-Related HCC

The 145 RNA-sequencing expression profiles and corresponding clinical information for HBV-related HCC were downloaded from the TCGA dataset. The results of overall survival analysis indicated that the expression levels of FANCD2 (*p* = 0.000193, HR = 2.689), CS (*p* = 0.00747, HR = 1.978) and SLC1A5 (*p* = 0.000123, HR = 2.738) affected the survival of HBV-related HCC patients, the survival advantage of low-expression groups was considerably greater as opposed to that of high-expression groups (Figure 2). The areas under the curve (AUC) of FANCD2 and SLC1A5 for one-, three-, and five-year overall survival (OS) were greater than 0.7 (Figure 2A,C). The disease-specific survival analysis of FANCD2 and SLC1A5 showed similar results (Appendix A).

### 3.3. SLC1A5 Combined with Clinicopathological Features of Nomogram Improves Prognosis and Survival Prediction of HBV-Related HCC

An effective nomogram model using the f HBV infection-associated FRGs and other clinicopathological information was constructed. Multivariate and univariate Cox regression analysis showed that SLC1A5 (*p* = 0.00111) was an independent prognostic indicator of HBV-related HCC (Figure 3A,B). Furthermore, a nomogram including SLC1A5 and several other clinical factors (age and T stage) was established to anticipate OS of HBV-related HCC patients. The nomogram showed that SLC1A5 (*p* < 0.001, C-index = 0.731) made a great contribution to the survival prediction (Figure 3C). The calibration curve illustrated that the anticipated probabilities of nomogram’s one-, three-, and five-year OS were close to the actually observed probabilities (Figure 3D).

### 3.4. Comparison of Immune Characteristics and Potential ICB Response Using SLC1A5

To study the immunological differences between the different expression level groups of SLC1A5 in HBV-related HCC, the relative abundance of 22 different types of immune cells was computed utilizing CIBERSORT. The results revealed that 8 types of immune cells showed significantly different estimated proportions in the high-SLC1A5 and low-SLC1A5 groups, with the proportion of T regulatory cells, macrophages and mast cells (*p* < 0.001) enriched in the high-SCL1A5 group (Figure 4A). The expression level of SLC1A5 was positively correlated with monocytes (Pearson correlation coefficient = 0.31), macrophage M1 (Pearson correlation coefficient = 0.5), CD8^+^ T cells (Pearson correlation coefficient = 0.4) and T regulatory cells (Pearson correlation coefficient = 0.35) (Figure 4B). 

To determine whether SLC1A5 can predict the response of HBV-related HCC patients to immune checkpoint inhibitor therapies, the expression levels of 8 immune checkpoint-related genes were analyzed in high-SLC1A5 and low-SLC1A5 HBV-related HCC patients. The levels of 6 immune checkpoint-related genes (CTLA4, HAVCR2, LAG3, PDCD1, PDCD1LG2 and TIGIT) were found to be significantly higher in the high-SLC1A5 group compared to the low-SLC1A5 group (Figure 4C). The high-SLC1A5 patients had a greatly elevated TIDE score as opposed to that of the low-SLC1A5 patients in the TCGA HBV-related HCC cohort, indicating that a great trend towards immune escape was observed in the high-SLC1A5 patient group, which may fail to respond to ICB treatment (Figure 4D).

### 3.5. The Correlations between SLC1A5 and Signaling Pathways in HBV-Related HCC

Further, a Spearman correlation analysis using TCGA data sets was performed to predict related signaling pathways of SLC1A5 in HBV-related HCC. 101 signaling pathways were analyzed, and SLC1A5 was positively correlated with 24 signaling pathways (Pearson correlation coefficient >0.3) and negatively correlated with 26 signaling pathways (Pearson correlation coefficient <−0.3) (Appendix A). As shown in Figure 5, SLC1A5 was tightly associated with apoptosis, cellular response to hypoxia, degradation of ECM, EMT, G2M checkpoint, IL-10 anti-inflammatory signaling pathway, inflammatory response, P53 pathway, PI3K-AKT-mTOR pathway and tumor proliferation signature.

## 4. Discussion

In recent years, with the increase in understanding of ferroptosis, the important functions have been extensively elucidated in HCC. Identifying regulators of ferroptosis would benefit researchers to clarify the mechanisms of ferroptosis in HBV-related HCC. In this study, four FRGs (FANCD2, CS, CISD1 and SLC1A5) were defined associated with HBV-related HCC using data from the TCGA database. In addition, we discovered that SLC1A5 was up-regulated in HBV-related HCC patients, and was negatively correlated with overall survival and disease-specific survival, and positively correlated with tumor progression. The data also showed that SLC1A5 was an independent risk factor for HBV-related HCC, and could increase the infiltrating levels of Treg and macrophage cells. Furthermore, the levels of six immune checkpoint-related genes (*CTLA4*, *HAVCR2*, *LAG3*, *PDCD1*, *PDCD1LG2* and *TIGIT)* were up-regulated in high-SLC1A5 patients. Accordingly, we proposed that SLC1A5 could play an important role in the progression and clinical immune therapy of HBV-related HCC. 

SLC1A5, a glutamine transporter on the cell membrane, has been found to positively modulate ferroptosis by increasing glutamine uptake and facilitating the generation of α-ketoglutarate generation to promote the progression of ferroptosis [14,21]. As a result, SLC1A5 suppresses tumor growth by supporting ferroptosis [22,23]. However, as a glutamine transporter, SLC1A5 also elevates glutamine consumption, which is a critical metabolic hallmark of tumors [24]. Because SLC1A5 is a double-edged sword in cancer progression, it is important to unravel its function in different cancers. It has been demonstrated that high SLC1A5 expression has been correlated with poor prognosis in many cancers, including hepatocellular carcinoma [25], lung cancer [14], breast cancer [26], head and neck squamous cell carcinoma [27], glioma [28] and pancreatic adenocarcinoma [29]. In HCC patients, SLC1A5 expression is significantly elevated in tumor tissues, compared with corresponding normal tissues [25]. High SLC1A5 expression is associated with poor overall survival, as well as increased numbers of tumor-infiltrating B cells, CD4^+^ T and CD8^+^ T cells, macrophages, neutrophils and dendritic cells [25]. These results were similar to ours and support that SLC1A5 regulates the tumor immune microenvironment to impact the efficacy of immunotherapy. In this work, we discovered that the high expression level of SLC1A5 was associated with poor overall survival, poor disease specific survival, tumor progression and immunosuppression. Additionally, consistent with Jewell’s study, this study also found that SLC1A5 regulated glutamine transport on the cell membrane, promoting mTORC1 translocation by Rag GTPase-dependent and -independent mechanisms to influence the PI3K-AKT-mTOR pathway [30]. These findings point toward the conclusion that SLC1A5 could facilitate the progression of HBV-related HCC via ferroptosis or the PI3K-AKT-mTOR pathway.

The immune microenvironment, consisting of immune cells and immune related molecules, acts as a crucial orchestrator of virus infection and tumor progression [31,32]. Recently, ferroptosis was discovered to be closely related to the immune microenvironment [33]. Ferroptotic cells may release lipid mediators to help recruit antigen presenting cells (APCs) and other immune cells to the ferroptotic tumor cells microenvironment [33]. In addition, deleting GPX4 in Treg cells, a key gene in ferroptosis, can lead to excessive accumulation of LPOs and ferroptosis, which promotes IL-1β production to enhance Th17 cell antitumor immune response [34]. However, it was intriguing to discover that some cells undergoing ferroptosis in TME may suppress the immune response. High levels of ROS inhibit T cell activation and proliferation and suppress the formation of TCR and MHC antigen complexes in T cells, thus inhibiting immune responses [35]. In addition, many immune cells are sensitive to ferroptosis; for example, significant lipid peroxidation occurs in CD36-positive CD8^+^ T cells, which results in ferroptosis and reduces the release of IFNγ to induce immunosuppression [36,37]. The level of glutamine in cytoplasm and the microenvironment is associated with immune cell responses [38]. In glioma cells, SLC1A5 could affect the infiltration and polarization of immune tumor-associated macrophages [28]. Combining an SLC1A5 inhibitor with an immune checkpoint inhibitor (ICI) can relieve immunosuppression and inhibit tumor growth [28]. Similarly, SLC1A5 expression is associated with tumor-infiltrating immune cells in the TME of stomach adenocarcinoma [39]. The SLC1A5 expression was negatively correlated with PD-L1 expression and positively correlated with PD-1 expression [39]. Meanwhile, in pancreatic adenocarcinoma, high SLC1A5 expression can reduce the infiltrating levels of CD8^+^ T cells, and is negatively correlated with the immune enrichment of CD8^+^ T cells, cytotoxic cells, NK cells and CD4^+^ T cells [29]. Thus, SLC1A5 plays an inhibitory effect on the antitumor immune process in pancreatic adenocarcinoma [29]. Due to the complex and variable tumor immune microenvironment in HCC, it is necessary to classify the function of ferroptosis in HCC to discover effective immunotherapy [40]. In this study, we demonstrated that SLC1A5, as an important modulator of ferroptosis, affects tumor immune microenvironment in HBV-related HCC via increasing the infiltrating Treg and macrophage cells, as well as up-regulating the levels of immune checkpoint-related genes (CTLA4, HAVCR2, LAG3, PDCD1, PDCD1LG2 and TIGIT). Tumor-associated macrophages play a key role in creating an immunosuppressive tumor microenvironment by producing cytokines, chemokines and growth factors, and trigger the release of inhibitory immune checkpoint proteins in T cells [41]. On the other hand, Tregs accumulate aberrantly in tumors to suppress antitumor immunity and support the establishment of an immunosuppressive microenvironment [42,43,44]. Additionally, immune checkpoint genes, such as PDCD1, CTLA4, LAG3, etc., regulate the T-cell exhaustion in HCC, impairing the T-cell capacity to secrete cytokines and proliferate [45]. Recently, the combination of atezolizumab (anti programmed death-ligand 1 [PD-L1]) and bevacizumab (anti–vascular endothelial growth factor [VEGF]) combination was approved as a first-line treatment for advanced HCC with superiority for sorafenib [46]. Thus, SLC1A5 plays a crucial role in the HCC immune microenvironment and contributes to the development and progression of HBV-related HCC. Taken together, understanding the overall characteristics of the HCC TME is essential for the design of novel combination therapies that inhibit tumorigenesis and/or restore sensitivity to immunotherapy-resistant tumors.

## 5. Conclusions

In the present study, elevated SLC1A5 was found to be an independent prognostic biomarker in patients with HBV-related HCC. High levels of SLC1A5 were associated with poor overall survival, poor disease specific survival and tumor progression. In addition, SLC1A5 may play an important role in the microenvironment of HBV-related HCC by regulating the infiltration of immune cells and resulting in an immunosuppressive microenvironment. Moreover, SLC1A5-induced high expression of immune checkpoint genes in HBV-related HCC may inhibit the therapeutic response of patients treated with ICIs. Thus, our findings provide new insights to assist clinicians develop appropriate therapeutic strategies and improve the long-term prognosis of HBV-related HCC.

## Figures and Tables

**Figure 1 jcm-12-01715-f001:**
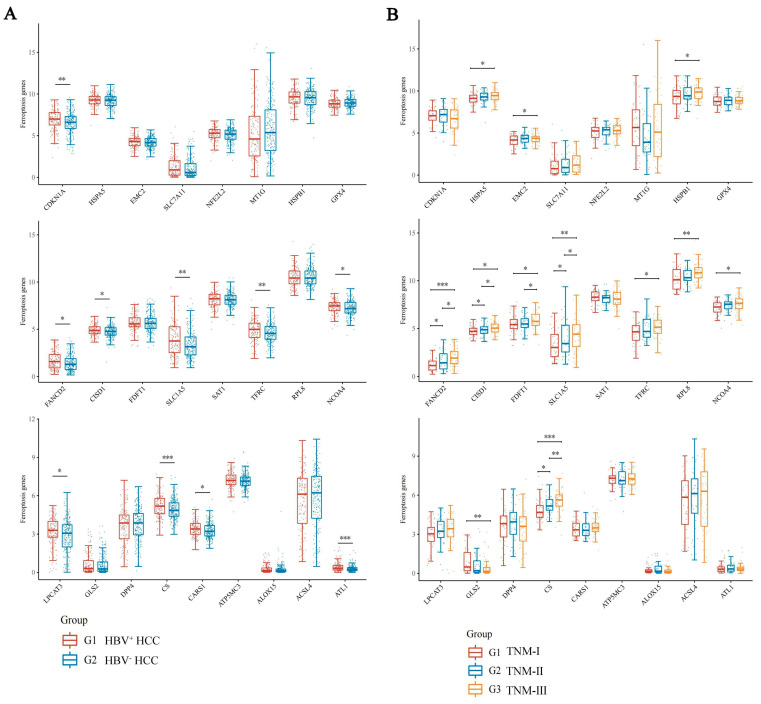
Expression distribution of ferroptosis-related gene mRNA in HCC tissues. The *x*-axis represents different ferroptosis related genes, and the *y*-axis represents gene expression distribution. Different colors represent different groups. (**A**) Expression distribution of ferroptosis related gene mRNA in HBV-positive HCC (red) and the HBV-negative HCC (blue). (**B**) The expression distribution of ferroptosis-related gene mRNA in HBV-related HCC with TNM I (red), II (blue) and III (yellow). * *p*  <  0.05, ** *p*  <  0.01, *** *p*  <  0.001. The statistical difference between two groups was compared through the Wilcox test, significance difference of three groups was tested with Kruskal–Wallis test.

**Figure 2 jcm-12-01715-f002:**
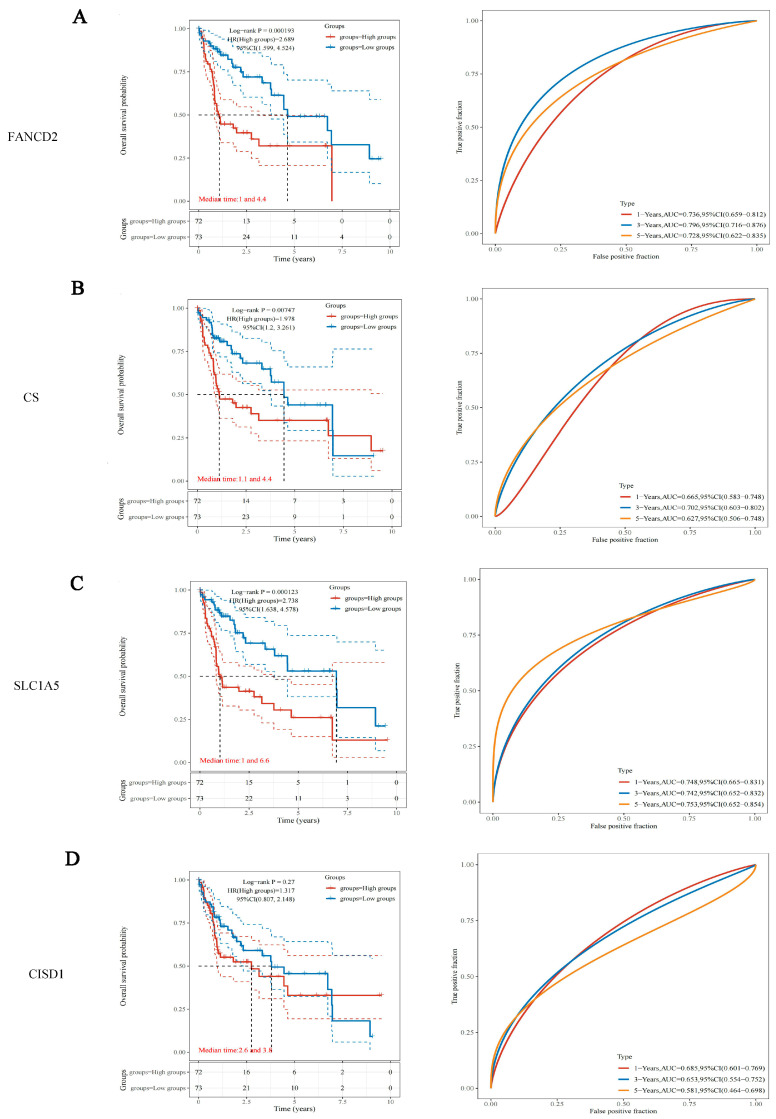
Overall survival analysis of FANCD2, CS, SLC1A5 and CISD1 in HBV-related HCC. (**left**) Kaplan–Meier survival analysis of FANCD2 (**A**), CS (**B**), SLC1A5 (**C**) and CISD1 (**D**) signature from the TCGA dataset, comparison among different groups was made by log-rank test. HR (High exp) represents the hazard ratio of the low-expression sample relatives to the high-expression sample. HR > 1 indicates the gene is a risk factor, and HR < 1 indicates the gene is a protective factor. HR(95%Cl), the median survival time (LT50) for different groups. (**right**) The ROC curve of the genes. Higher values of AUC correspond to higher predictive power.

**Figure 3 jcm-12-01715-f003:**
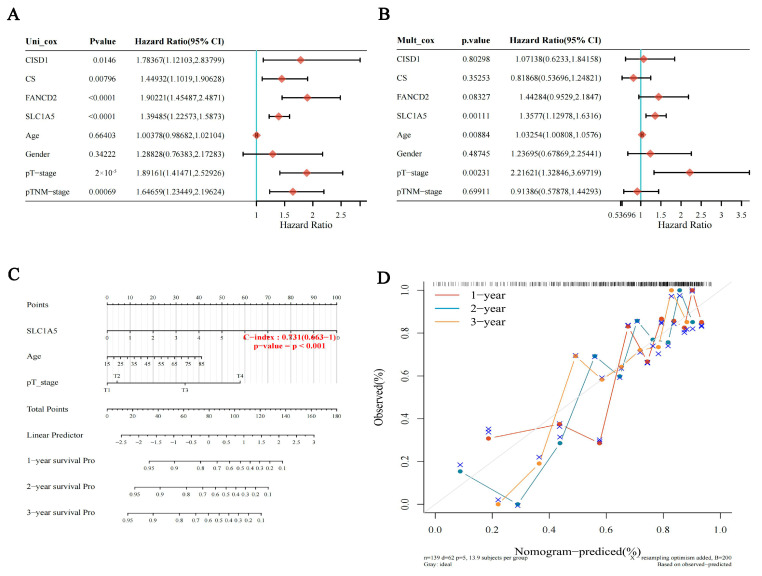
Analysis of prognostic value for FANCD2, CS, SLC1A5 and CISD1 in HBV-related HCC. The *p* value, risk coefficient (HR) and confidence interval were analyzed by univariate (**A**) and multivariate (**B**) Cox regression. (**C**) Nomogram can predict the 1-yea, 2-year and 3-year overall survival of HBV-related HCC patients. (**D**) Calibration curve for the overall survival nomogram model. The dashed diagonal line represents the ideal nomogram, and the blue line, red line and orange line represent the 1-year, 2-year and 3-year of the observed nomogram.

**Figure 4 jcm-12-01715-f004:**
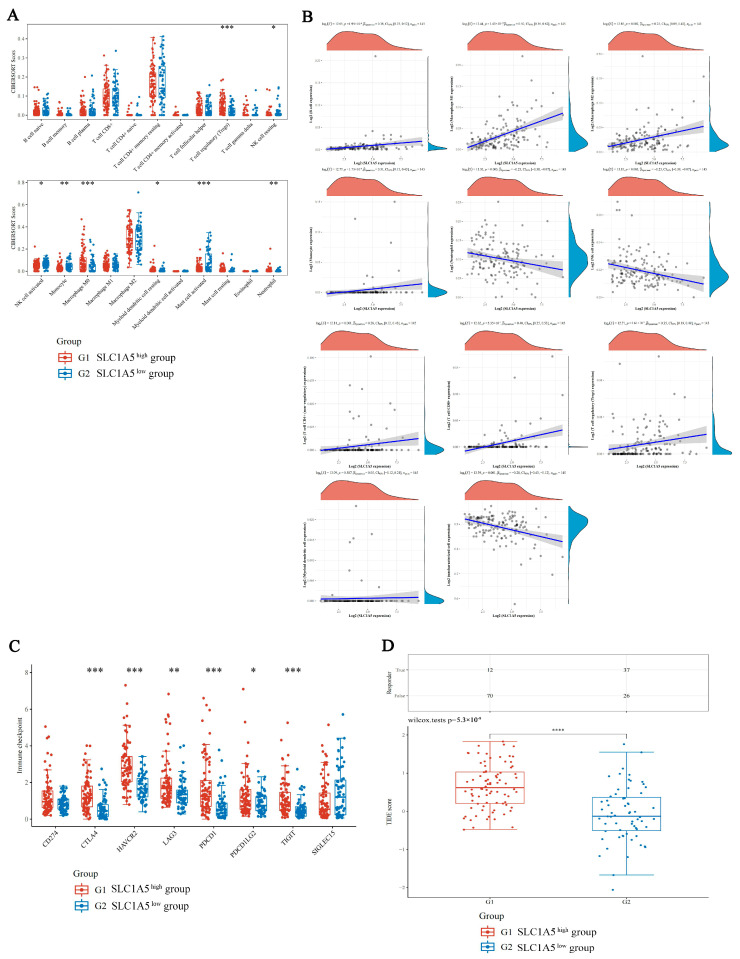
Comparison of immune characteristics and potential ICB response in different expression level groups of SLC1A5 in HBV-related HCC. (**A**) The expression distribution of CIBERSORT immune score in high-SLC1A5 (red) and low-SLC1A5 (blue) groups. The *x*-axis represents immune cell types, and the *y*-axis represents the expression distribution of immune score in different groups. (**B**) The correlations between SLC1A5 expression and immune score were analyzed with Spearman. The *x*-axis represents the distribution of the SLC1A5 expression or the score, and the *y*-axis represents the distribution of the immune score. The density curve on the right represents the trend in the distribution of the immune score, and the upper density curve represents the trend in the distribution of the gene expression or the score. The value above represents the correlation *p* value, correlation coefficient and correlation calculation method. (**C**) The expression distribution of immune checkpoints gene in high-SLC1A5 (red) and low-SLC1A5 (blue) groups. The *x*-axis represents different groups of samples, and the *y*-axis represents the expression distribution of gene. (**D**) Statistical table of immune response of samples and the distribution of immune response scores in high-SLC1A5 (red) and low-SLC1A5 (blue) groups. * *p*  <  0.05, ** *p*  <  0.01, *** *p*  <  0.001, **** *p*  <  0.0001.

**Figure 5 jcm-12-01715-f005:**
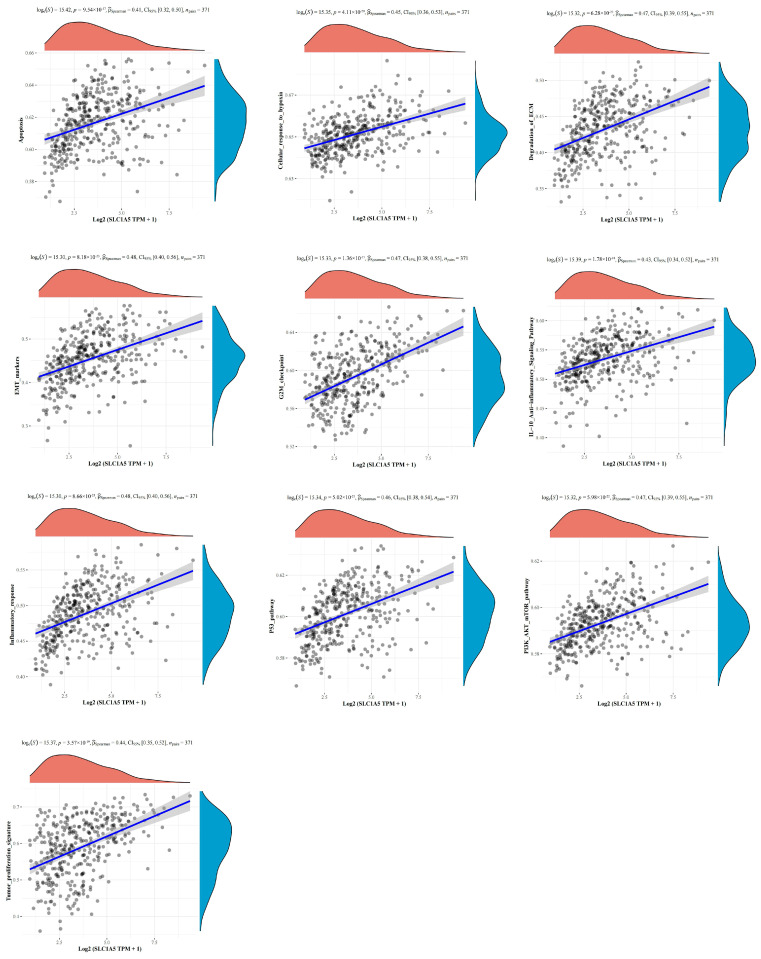
The correlations between the expression of SLC1A5 and pathway score analysis with Spearman. The *x*-axis represents the distribution of SLC1A5 expression and the *y*-axis represents the distribution of the pathway score. The density curve on the right represents the trend in distribution of pathway immune score, and the upper density curve represents the trend in distribution of gene expression. The values on top show the correlation *p* value, correlation coefficient and correlation calculation method.

## Data Availability

The datasets generated and/or analyzed during the current study are available from the corresponding author upon reasonable request.

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
