# Peer review of "Ferroptosis-Related Gene SLC1A5 Is a Novel Prognostic Biomarker and Correlates with Immune Microenvironment in HBV-Related HCC"

_jcm, 2023, doi:10.3390/jcm12051715_

Round 1
Reviewer 1 Report
In the article “Ferroptosis-related gene SLC1A5 is a novel prognostic biomarker and correlates with immune microenvironment in HBV-related HCC” by Su et al., the authors have proposed SLC1A5 as a new prognostic marker for HBV-related HCC. The authors have performed meta-analysis of HBV-related HCC (HBV+ and HBV-) dataset available on TCGA to identify FRG signatures. They have further performed CIBERSORT and TIDE-based analyses to assess the role of immune regulatory mechanisms pertinent to the prognosis and pathogenesis of HBV-related HCC in the context of SLC1A5 expression. While the authors have utilized a multipronged approach to identify a new prognostic biomarker for patients with HBV-related HCC, the study design and rationale are not clear. In addition, studies by Zhao et al. (34367941) show similar results which reduce the novelty of this study. The manuscript may be considered the following concerns are addressed.
Major concerns:
1. While the authors have chosen the four shortlisted genes as FRG signatures based on their significance in Figures 1A and 1B, the authors should also investigate LPCAT3 which shows a significant increase in HBV+ HCC patients and a stage-specific increase in expression.
2. The resolution of the figures is very poor, and the text is not easily readable or unreadable in many instances to draw any conclusions. The authors must include high-resolution images in the revised version of the manuscript.
3. The authors must include the disease-specific survival probability of CS and CISD1 as well.
4. It is not clear why the authors have assessed the immunological characteristics. The authors must include the rationale for evaluating the immune aspect of the study in the results section.
5. It is not clear if SLC1A5 is correlated to pro- or anti-inflammatory response as the expression of SLC1A5 is positively correlated to M1 macrophages and CD8+ T cells which are anti-tumorigenic. Can the authors elaborate on their conclusions from the CIBERSORT analyses?
6. The authors must include a table collating the list of positively and negatively associated signaling pathways with SLC1A5 expression to derive any conclusions/significance of the regulated pathways.
7. The authors have only focused on the role of SLC1A5 in the later part of the study. However, the FANCD2 gene seems to be a prospective prognostic marker and the authors must include the analysis for FANCD2 as supplementary figures.
Minor concerns:
1. In figure 1B, the statistical comparisons are not very clear. The authors should include different symbols to denote the statistical analysis between TNM stages 1 and 2, and TNM stages 1 and 3.
Author Response
Dear Reviewer,
We have submitted a revised version of the manuscript (Title: Ferroptosis-related gene SLC1A5 is a novel prognostic biomarker and correlates with immune microenvironment in HBV-related HCC) to address the points raised during the review process. Revisions in the manuscript were marked in red.
Find below a point-by-point response to your valuable comments.
Thank you very much for your valuable comments and suggestions, which will certainly help to improve the quality of our work.
Major concerns
Comment 1: While the authors have chosen the four shortlisted genes as FRG signatures based on their significance in Figures 1A and 1B, the authors should also investigate LPCAT3 which shows a significant increase in HBV+ HCC patients and a stage-specific increase in expression.
Response 1: Thank you for your valuable feedback. We agree with the reviewer that LPCAT3 is an interesting gene, which shows a significant increase in HBV+ HCC patients and a stage-specific increase in expression. However, as shown in revised Fig.1, LPCAT3 did not show significant differences among the stage-specific three groups.
Comment 2: The resolution of the figures is very poor, and the text is not easily readable or unreadable in many instances to draw any conclusions. The authors must include high-resolution images in the revised version of the manuscript.
Response 2: Thank you for bringing this issue to our attention. We apologize for the poor resolution of the figures and any inconvenience it may have caused. We have provided high-resolution images in the revised version of the manuscript to ensure that the text is easily readable and conclusions can be drawn from the figures. Besides, the high-resolution images with 600 dpi have been submitted to the journal. We appreciate the reviewer's suggestion and will take all necessary steps to ensure that the revised manuscript meets the high standards of the journal.
Comment 3: The authors must include the disease-specific survival probability of CS and CISD1 as well.
Response 3: Thank you for your preciseness. The disease-specific survival probability of CS and CISD1 have been added to Supplemental Fig. 1 of the revised manuscript. We appreciate the reviewer's feedback and believe that this addition will improve the scientific rigor and overall impact of our study.
Comment 4: It is not clear why the authors have assessed the immunological characteristics. The authors must include the rationale for evaluating the immune aspect of the study in the results section.
Response 4: Thank you for pointing that out. Indeed, it would be important to clarify the rationale for evaluating the immune aspect of the study in the Results section. A brief explanation was added in line 291 of the revised manuscript, “Immune microenvironment, consist of immune cells and immune related molecules, act as crucial orchestrators of virus infection and tumor progression [doi: 10.1016/j.jhep. 2021.08.029, doi: 10.1016/j.jhepr.2021.100388
]. Recently, ferroptosis was discovered closely related to the immune microenvironment [doi:10.1038/s41568-019-0149-1
].” Therefore, the immunological characteristics of SLC1A5 were assessed to determine its impact on the immune microenvironment.
Comment 5: It is not clear if SLC1A5 is correlated to pro- or anti-inflammatory response as the expression of SLC1A5 is positively correlated to M1 macrophages and CD8+ T cells which are anti-tumorigenic. Can the authors elaborate on their conclusions from the CIBERSORT analyses?
Response 5: Thank you for taking the time to review our manuscript and provide valuable feedback. By using CIBERSORT analysis, we discovered T regulatory cell, macrophage and mast cell (P < 0.001) enriched in the high-SCL1A5 group (Fig. 4A). Tregs are believed to play a negative role during immune surveillance, resulting in tumor tolerance. HBV infection in particular facilitates the recruitment and accumulation of massive numbers of Tregs into the TME, which hampers effective antitumor responses and contributes to poor prognosis. The correlation analysis using the TCGA database showed that SLC1A5 was positively correlated with monocytes, macrophage M1, CD8+ T cells and T regulatory cells. Due to the complexity of the immune microenvironment, this data cannot provide a clear indication of the specific role of SLC1A5 in the immune microenvironment. We will conduct further research on the role of SLC1A5 in the immune microenvironment in future studies.
Comment 6: The authors must include a table collating the list of positively and negatively associated signaling pathways with SLC1A5 expression to derive any conclusions/significance of the regulated pathways.
Response 6: Thank you for your suggestion. We added a supplementary table collating the list of positively and negatively associated signaling pathways with SLC1A5 expression in the revised version of the manuscript.
Comment 7: The authors have only focused on the role of SLC1A5 in the later part of the study. However, the FANCD2 gene seems to be a prospective prognostic marker and the authors must include the analysis for FANCD2 as supplementary figures.
Response 7: Thanks for your professional advice. Multivariate Cox regression analysis showed that FANCD2 (P = 0.08327) was not an independent prognostic indicator of HBV-related HCC. So we did not focus on the role of FANCD2 in the later part of the study. In the revised version of the manuscript, the comparison of immune characteristics and potential ICB response in different expression level groups of FANCD2 in HBV-related HCC was shown in Supplemental Fig. 2. We appreciate your valuable input and will do our best to improve the manuscript accordingly.
Minor concerns
Comment 1: In figure 1B, the statistical comparisons are not very clear. The authors should include different symbols to denote the statistical analysis between TNM stages 1 and 2, and TNM stages 1 and 3.
Response 1: Thank you for your valuable suggestion. We appreciate your feedback and revise Fig. 1B to include different symbols to denote the statistical comparisons between TNM stages 1 and 2, TNM stages 1 and 3, and TNM stages 2 and 3. We believe that this will improve the clarity and understandability of the figure.
Reviewer 2 Report
In the manuscript „ Ferroptosis-related gene SLC1A5 is a novel prognostic biomarker and correlates with immune microenvironment in HBV-related HCC " by Su et al., the authors identified Ferroptosis-related gene SLC1A5 as a prognosis marker for hepatocellular cancer. Thus, SLC1A5 could be used in supplement to other clinical biomarkers for early cancer diagnosis. The study is well performed.
I would like to suggest:
- Providing basic information on the protein itself in the introduction
- Fig 1, In the figures for some genes the authors have on the top the symbol “-“ when there is no statistical difference between two groups but for others there is nothing. Please uniformize.
- In the results section the phrase “…made a great contribution to the survival prediction of nomogram” needs to be changed for clarification.
- The quality of the English language should be strictly revised.
Author Response
Dear Reviewer,
We have submitted a revised version of the manuscript (Title: Ferroptosis-related gene SLC1A5 is a novel prognostic biomarker and correlates with immune microenvironment in HBV-related HCC) to address the points raised during the review process. Revisions in the manuscript were marked in red.
Find below a point-by-point response to your valuable comments.
Thank you very much for your friendly evaluation of the manuscript, constructive comments and valuable suggestions to improve the quality of our work.
Major concerns
Comment 1: Providing basic information on the protein itself in the introduction.
Response 1: Thank you for your feedback. We appreciate your suggestion to provide basic information on the protein itself in the introduction. We revise the introduction section to include a brief overview of SLC1A5 in line 60 of the revised version of the manuscript. Thank you for bringing this to our attention, and we will make the necessary changes to improve the clarity and completeness of the manuscript.
Comment 2: Fig 1, In the figures for some genes the authors have on the top the symbol “-“ when there is no statistical difference between two groups but for others there is nothing. Please uniformize.
Response 2: Thank you for your comment. We apologize for the lack of consistency in the notation used in Fig 1. In the revised version, we will make sure to uniformize the symbols used throughout the figures and provide clear legends to explain the meaning of each symbol.
Comment 3: In the results section the phrase “…made a great contribution to the survival prediction of nomogram” needs to be changed for clarification.
Response 3: Thank you for your suggestion. We apologize for any confusion caused by the unclear phrase in our manuscript. We have revised the sentence to provide a clear explanation of the contribution of SLC1A5.
Comment 4: The quality of the English language should be strictly revised.
Response 4: We appreciate your comment and apologize for any inconvenience caused by the quality of the English language in the manuscript. We will perform a thorough language revision to ensure that the revised version of the manuscript is of the highest quality.
Author Response
Dear Reviewer,
We have submitted a revised version of the manuscript (Title: Ferroptosis-related gene SLC1A5 is a novel prognostic biomarker and correlates with immune microenvironment in HBV-related HCC) to address the points raised during the review process. Revisions in the manuscript were marked in red.
Find below a point-by-point response to your valuable comments.
Thank you very much for your friendly evaluation of the manuscript, constructive comments and valuable suggestions to improve the quality of our work.
Major concerns
Comment 1: The authors provide a sufficient background of the research and an adequate introduction, however, the mechanism of ferroptosis should be further discussed, as well as its interplay with tumor microenviroment. Similarly, SLC1A5 role should also be discussed in the introduction.
Response 1: Thank you for your valuable feedback. We incorporated a more detailed introduction of the mechanism of ferroptosis and the role of SLC1A5 in line 60 of the revised version of the manuscript. We appreciate your suggestion and believe that these revisions will strengthen the manuscript.
Comment 2: Additional recent references relevant to the research should be added.
Response 2: Thank you for your comment. We reviewed the literature again and added the most relevant and recent references to the manuscript in the revised version. Thank you for your helpful feedback, and we will ensure that the updated manuscript reflects the current state of research in the field.
Comment 3: English language and style are fine; however minor spell check should be performed.
Response 3: Thank you for bringing this to our attention. We perform a thorough spell check to ensure that the manuscript is free of any spelling errors, and make the necessary corrections to improve the quality of the manuscript in the revised version.
Round 2
Reviewer 1 Report
The authors have addressed the previously raised questions in the revised version of the manuscript. The manuscript may be accepted in its present form and published in the journal.